# Supporting adolescents living with HIV within boarding schools in Kenya

**Judith Kose[1,2¤]\*, Cosima Lenz[3], Job Akuno[4], Fred Kiiru[5], Justine Jelagat Odionyi[4], Rose Otieno-Masaba[4], Elizabeth A. Okoth[4], Godfrey Woelk[6], Solomon Leselewa[7], Pieter Leendert Fraaij[8], Natella Rakhmanina[3,9,10]**

**1** Technical Strategy and Innovation, The Elizabeth Glaser Pediatric AIDS Foundation, Nairobi, Kenya, **2** Erasmus MC, Department of Viroscience, Erasmus University, Rotterdam, the Netherlands, **3** Technical Strategy and Innovation, The Elizabeth Glaser Pediatric AIDS Foundation, Washington, DC, United States of America, **4** Country Program, The Elizabeth Glaser Pediatric AIDS Foundation, Nairobi, Kenya, **5** Ministry of Education, Homa Bay County, Kenya, **6** Research Department, The Elizabeth Glaser Pediatric AIDS Foundation, Washington, DC, United States of America, **7** Teachers Service Commission, Homa Bay County, Kenya, **8** Pediatric Infectious Diseases Division, Erasmus MC-Sophia/ Erasmus University, Rotterdam, the Netherlands, **9** The George Washington University, Washington, DC, United States of America, **10** Children's National Hospital, Washington, DC, United States of America

¤ Current address: Ariel House, Westlands, Nairobi, Kenya
\* jkose@pedaids.org

**Data Availability Statement:** All relevant data within the manuscript and its Supporting information files contain the minimal data set.

## Abstract

Adolescents and youth living with HIV (AYLHIV) are a uniquely vulnerable population facing challenges around adherence, disclosure of HIV status and stigma. Providing school-based support for AYLHIV offers an opportunity to optimize their health and wellbeing. The purpose of this study was to evaluate the feasibility of school-based supportive interventions for AYLHIV in Kenya. From 2016–2019, with funding from ViiV Healthcare, the Elizabeth Glaser Pediatric AIDS Foundation implemented the innovative Red Carpet Program (RCP) for AYLHIV in participating public healthcare facilities and boarding schools in Homa Bay and Turkana Counties in Kenya. In this analysis, we report the implementation of the school-based interventions for AYLHIV in schools, which included: a) capacity building for overall in-school HIV, stigma and sexual and reproductive health education; b) HIV care and treatment support; c) bi-directional linkages with healthcare facilities; and d) psychosocial support (PSS). Overall, 561 school staff and 476 school adolescent health advocates received training to facilitate supportive environments for AYLHIV and school-wide education on HIV, stigma, and sexual and reproductive health. All 87 boarding schools inter-linked to 66 regional healthcare facilities to support care and treatment of AYLHIV. Across all RCP schools, 546 AYLHIV had their HIV status disclosed to school staff and received supportive care within schools, including treatment literacy and adherence counselling, confidential storage and access to HIV medications. School-based interventions to optimize care and treatment support for AYLHIV are feasible and contribute to advancing sexual and reproductive health within schools.

**Funding:** This work was supported with funding by a grant from the Positive Action for Adolescents Programme by ViiV Healthcare UK Ltd, London, UK under the Positive Action for Adolescents Program. EGPAF's adolescent and youth-focused programs in Kenya are supported by funding from United States Government (USG), ELMA Philanthropies, and ViiV Healthcare UK Ltd, London, UK. The funder did not participate in the implementation of the program or interpretation of the results as presented in this manuscript. The results do not necessarily represent the views of the funder. The funders had no role in study design, data collection and analysis, decision to publish, or preparation of the manuscript.

**Competing interests:** This work was supported with funding by a grant from the Positive Action for Adolescents Programme by ViiV Healthcare UK Ltd. There are no patents, products in development or marketed products to declare. This does not alter our adherence to PLOS ONE policies on sharing data and materials. The authors have declared that no competing interests exist.

## Introduction

School attendance is a critical component in the lives and development of adolescents and youth, including those living with HIV [1–3]. School experience contributes to the emotional, social, cognitive, and behavioral development of children and adolescents [1], and provides a unique opportunity to reach adolescents and youth living with HIV (AYLHIV) beyond their homes to support their health, treatment, self-management and wellbeing [4, 5].

Public day and boarding schools are two of the most common forms of academic settings for children and youth in many countries, including Kenya [6]. Data from the Kenya Ministry of Education reports that there are approximately 4,300 secondary boarding schools nationally, which constitute 50% of all public secondary schools [7]. In Kenya, adolescents and youth spend 8–10 hours per week in a day school and 9 months of their daily lives in a boarding school during the school year, where they reside away from their homes [8]. For youth at boarding schools, their teachers, school staff, and peers become their main community and source of support as daily contact with families and caregivers is limited [9].

The boarding school environment poses specific challenges to AYLHIV [10–12]. Life at a boarding school affects access to HIV care and treatment compared to AYLHIV who attend day schools and have their families' support and greater flexibility around healthcare facility access. Barriers and challenges to HIV care in boarding schools include restricted independence and increased controlled access to healthcare facilities (HCFs) and external providers, HIV status disclosure challenges, limited adherence support and difficulties with accessing daily antiretroviral treatment (ART), self and experienced stigma, and limited privacy and confidentiality [10, 13–17]. There is an unmet need to design and evaluate interventions to address these challenges and improve health outcomes among AYLHIV, especially those attending boarding schools [18, 19].

Kenya carries a high HIV burden among adolescents and youth with an estimated 91,634 adolescents aged 10–19 years and 145,471 youth aged 15–24 years living with HIV in 2019 [20]. In response to the needs of AYLHIV, the Elizabeth Glaser Pediatric AIDS Foundation's (EGPAF) Red Carpet program (RCP), funded by ViiV Healthcare Positive Action, implemented, three packages of interventions including a healthcare facility intervention package, a meaningful youth engagement package and boarding school-based intervention package in Homa Bay and Turkana Counties, two high HIV epidemic burden regions in Kenya [21]. The main goal of the school-based RCP interventions was to provide care and treatment support for AYLHIV aged 10–24 years. Implemented in coordination with the Kenya Ministry of Education (MOE) and Ministry of Health (MOH), the RCP school interventions also included building of school capacity in sexual and reproductive health, HIV and stigma education and strengthening bi-directional coordination between RCP boarding schools and HCFs. The aim of this analysis is to evaluate the feasibility of the school-based RCP interventions for AYLHIV at participating boarding schools in Kenya.

## Methods

### Study design

This study analyzed programmatic data from implementation of the RCP boarding school-based intervention package in Homa Bay and Turkana Counties in Kenya over three project years from March 2016 through September 2019.

The RCP study protocol was approved by the Kenyatta National Hospital-University of Nairobi Ethical Review Board, Kenya, and Chesapeake Institutional Review Board, USA. The study obtained a waiver of informed consent, as it used programmatic de-identified data and

posed no more than minimal risk to participants. No individual patient level data were collected during RCP evaluation.

## RCP boarding school selection and engagement

The process of involving schools into RCP began with engaging and sensitizing the regional MOH and MOE on RCP to ensure project ownership and support. In collaboration with county MOH, MOE and the Teachers Service Commission (TSC), RCP conducted a joint mapping exercise to identify the boarding schools attended by AYLHIV in care and located within the catchment area of local HCFs. Following identification of schools to be engaged in the project, a joint MOH and MOE meeting was conducted to establish sub-county working groups to lobby for access to schools in addition to coordination, planning, and implementation of the RCP school-based project. RCP team and sub-county working groups led the series of meetings with key sub-county stakeholders (TSC, Parents Association (PA), Parent Teachers Associations (PTA), School Health Committees (SHC) and AYLHIV champions) and heads of school to discuss implementation of RCP and the needs of AYLHIV. With affirmed participation, RCP schools and their respective SHC were linked to the local RCP HCFs and sub-county RCP teams for project implementation.

The main criteria for RCP HIV responsive schools were established by the EGPAF RCP team and included a checklist of target characteristics which were reviewed and approved by county and sub-county stakeholders (S1 Appendix). Based on these criteria, within each individual school, head teacher or principle, SHC members, school-based peer educators, sub-county RCP Adolescent Youth Peer Advisory Group (AYPAG) and healthcare workers (HCWs) from interlinked RCP HCFs facilities, jointly designed and further implemented individual school RCP activities. AYPAGs are comprise of trained AYLHIV who participate on a voluntary basis to represent the interests of AYLHIV in advocacy and implementation activities at the facility, sub-county, and county level in discussions and working groups, along with supporting peers at the facility level.

School-based RCP package interventions included: a) capacity building for overall in-school HIV, stigma and sexual and reproductive health (SRH) education; b) HIV care and treatment support; c) bi-directional linkages with HCFs; and d) psychosocial support (PSS) (Fig 1). Each school-based intervention comprised of various activities involving AYLHIV, HCWs, school staff, SCH members, adolescent and youth peer volunteers and newly identified and trained adolescent health advocates (AHA).

**Capacity building for overall in-school HIV, stigma and SRH education** included support for existing SHCs and AHA training, and school-wide SRH, HIV, and stigma education. In order to foster productive dialogues in schools, relevant, age appropriate, evidence-based information on HIV and SRH was provided and discussed in various venues including school health days and school gatherings. The school-wide provision of information and dialogues on HIV supported building understanding among school staff and adolescents and youth overall of discriminatory or stigmatizing behaviors towards people living with HIV.

SHCs already existed within the MOE policy framework regionally and nationwide; however, RCP aimed to ensure the integration of supportive elements for AYLHIV within SHC roles and responsibilities (Fig 2). SHCs at RCP schools required school staff members and/or representatives of the school to receive training on how to support AYLHIV in school settings. Training for teachers and school support personnel was conducted using an EGPAF developed curriculum adapted from the national Adolescent Package of Care (APOC) [22] training. RCP trainings were offered to various school staff including guidance counselling teachers, boarding masters/mistresses, school matrons, school nurses, and members of SHCs. RCP trainings

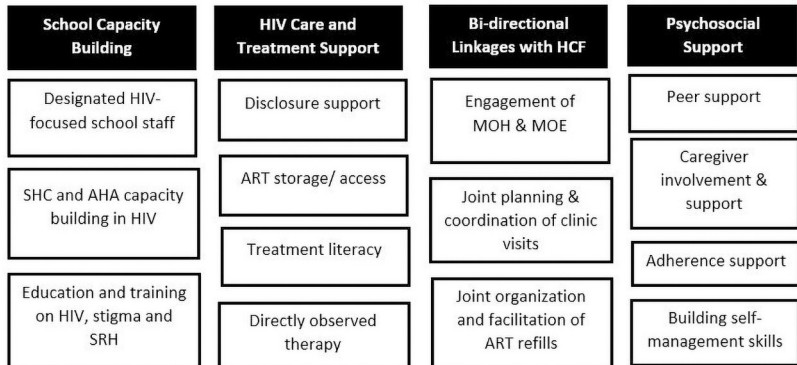

**Fig 1. Key interventions of RCP package in schools, Homa Bay and Turkana Counties, Kenya.**

emphasized privacy and confidentiality to assure trust and to prevent stigma and discrimination towards AYLHIV. In recognizing the significant challenge and barrier stigma and discrimination play in the school and treatment experience for AYLHIV [11, 16], RCP also engaged parents and students in education and practical skill building activities to empower the broader school community with the tools and knowledge to prevent and combat HIV stigma.

Trained school staff (mostly teachers) were asked to volunteer for the role of adolescent health advocates (AHA) to directly support RCP coordination and implementation in schools and to lead the education efforts on the needs of AYLHIV with a focus on support for care and treatment. These more specific education and trainings focused on sensitizing school personnel, SHC members and AHA on the needs of AYLHIV, including support for disclosure of HIV status, dealing with internal and external stigma, gaining treatment literacy, assuring adherence support and ARVs storage, and providing PSS. These trainings employed skill-building practice sessions on counseling, communication skills, and strategies to support AYLHIV.

Several designated focal contacts at schools (school staff, SHC members and AHA) were created and advertised across the school community to facilitate AYLHIV approaching these individuals without fear of repercussion for their needs, including disclosure support, gaining confidential access to the medication in storage or planning attendance clinic appointments.

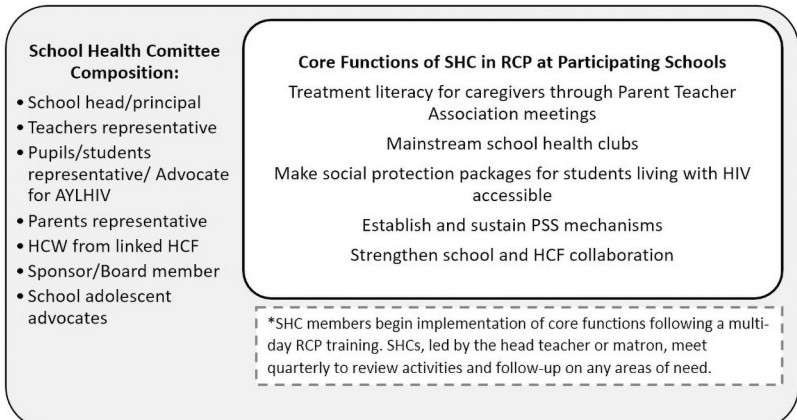

**Fig 2. School Health Committee (SHC) composition and core functions in RCP schools.**

**HIV care and treatment support** started with the process of disclosure of HIV status at schools among AYLHIV. The support for the disclosure of HIV status was offered to AYLHIV and/or their caregivers. AHA, SHC and designated school staff were trained on supporting two types of disclosure of HIV status: a) AYLHIV who knew their status were supported to disclose their status to the school staff, caregivers and peers, where applicable; and b) caregivers were supported to disclose the AYLHIV status to the school staff and to the AYLHIV who were unaware of their HIV status. Both types of disclosure resulted in school staff knowing the HIV status of AYLHIV in order to support their needs. SHC, AHA and school staff made the wider school community (including PA and students) aware that disclosure support was available and emphasized the confidentiality and discretion of the process. RCP school staff, SHC and AHA regularly reached out to the school community reaffirming the availability of support for disclosure of HIV status. Every effort was made to provide disclosure support in a way that would minimize accidental disclosure and prevent internal and external stigma. Once disclosed, AYLHIV were supported with treatment literacy education and treatment counselling, confidential access to ART, and bi-directional support in coordination of refills and appointments with HCFs. For AYLHIV who had their HIV status disclosed, storage of their ARVs in designated drug cabinets in private spaces was offered. AYLHIV had the option of morning visits to this private space where the school matron or nurse would assure their access to their medication. In order to minimize the potential stigma among peers, the space for ART storage and administration was shared with other adolescents and youth accessing medical services within the school for other chronic health conditions. For AYLHIV experiencing significant adherence challenges, directly observed therapy (DOT) was administered daily with guidance and in coordination with HCWs from linked HCFs in confidential and private settings by a school matron, school nurse, or trained AHA.

**Bi-directional linkages with HCFs** included joint planning and coordination of ART refills, AYLHIV clinical visits, and referrals to additional services. HCFs RCP coordinators communicated with designated AHA and school staff on AYLHIV needs for clinical follow up and refills, while school staff tracked AYLHIV attendance to clinic visits for treatment and follow-up and equally assured prompt communication with interlinked HCFs for ARV refills and any other clinical needs. Those AYLHIV who received care at HCFs outside of the RCP project areas, received school-based care, treatment support and PSS support.

**Psychosocial support services (PSS)** for AYLHIV included peer counselling and support conducted by AHA with engagement of peer facilitators and AYPAG members, focusing on positive living with HIV and development and/or enhancement of self-management skills. Peer counselling and support was available on a regular basis (bi-weekly) in private, confidential spaces via individual and/or group (minimum of two AYLHIV) sessions. When feasible and applicable, caregivers were invited to engage in treatment literacy sessions to support AYLHIV. Caregivers received treatment literacy and adherence counselling through the PTA, which included sensitization on AYLHIV needs, RCP and skills building around child-caregiver communication.

## Data analysis

Routine programmatic RCP data collected included: number of schools in the project; number of RCP HCFs linked to schools; number of school staff, AHA, and SHC staff trained; number of AYLHIV who had their HIV status disclosed at school; number of AYLHIV who accessed ART support and PSS at school; and estimated number of adolescents and youth in schools reached with HIV, SRH and stigma educational activities. The school-based care and treatment support and linkage between schools and HCFs were assessed through documenting the

**Table 1. Cumulative summary of the RCP achievements by project year, 2016–2019.**

| Project Year | Number of participating schools | Number of HCFs linked with schools | Number of AYLHIV supported | Number of AY in RCP schools | Number of school staff sensitized | Number of AHAs |
|---|---|---|---|---|---|---|
| 1 | 67 | 50 | 264 | 986 | 420 | 140 |
| 2 | 80 | 66 | 362 | 1028 | 325 | 300 |
| 3 | 87 | 66 | 546 | 1774 | 361 | 476 |

mutual referral/communication processes via point of contact staff assigned at HCFs and schools, documentation of the HCF within the school health records for the disclosed AYLHIV and documentation of school-based support (access to ART and PSS) within schools and HCF records. Programmatic monitoring and evaluation visits were used to verify the bi-directional connections between HCF and schools with random client reviews. The mean viral suppression rate among the AYLHIV (stratified by age) at RCP HCFs was calculated using viral suppression programmatic reports from the county HCFs. Descriptive statistics were used for data analysis. Since RCP was a programmatic intervention and not a prospective trial of new interventions, including a control group of non-intervention schools while being desirable was not feasible.

## Results

RCP achievements throughout the overall period of study implementation of the school-based package is summarized in Table 1. Following the pilot of RCP school-based support in 25 high volume secondary boarding schools in Homa Bay County, the number of participating secondary boarding schools increased gradually in Homa Bay Country in the first two years of the project, with expansion to Turkana County in the third project year reaching a total of 87 schools (74 in Homa Bay County and 13 in Turkana County).

A total of 561 school staff were trained over the project period. Overall, 476 AHA received capacity building training to support their roles in RCP participating schools. Based on school-wide engagement and documented school attendance, an estimated 1774 students were reached across participating schools with various educational events focusing on SRH, HIV and stigma through diverse platforms including school health days and school gatherings.

By the end of the third project year, all 87 RCP schools were inter-linked to 66 local HCFs to support the care and treatment of AYLHIV. Across the 87 schools, 546 AYLHIV (~ 6 AYLHIV per RCP school) had their HIV status disclosed at school and received care and treatment and PSS within schools; 536 AYLHIV received bi-directional coordination of their care with interlinked HCFs (10 AYLHIV had their care at non-RCP HCFs and received school-based support only).

While baseline data for the HIV treatment outcomes were not collected, the mean viral suppression rate among the AYLHIV at RCP HCFs was 82% during 2019, the third project year, with 87% among AYLHIV aged 10–14 years, 90% for AYLHIV aged 15–19 years and 93% for AYLHIV aged 20–24 years.

## Discussion

The implementation of RCP demonstrated the feasibility of integrating support for AYLHIV within boarding schools in high HIV burden areas in Kenya. AYLHIV have been reported to face multiple challenges with adherence, disclosure of HIV status and stigma within school environments [4, 23, 24]. The WHO acknowledges the critical opportunity in linking schools and community services to support an integrated approach to health along with the need for

capacitation and active engagement of teachers and school staff in school-based health programs [25]. To date, a limited number of studies have shown school's capacity to reduce barriers to care among AYLHIV by providing ART support, linkages to HCFs, and improving overall outcomes of HIV disease, such as viral suppression and living positively [14, 26].

Schools can play a significant role in improving the management and outcomes of several chronic diseases affecting adolescent populations. For instance, school-based asthma programs have demonstrated significant improvements in self-management, responsibility, coping with negative feelings, and health outcomes among children and youth [27, 28]. School-based programs addressing mental health conditions have also demonstrated reduction in mental health symptoms and improved function among youth and support the feasibility and efficacy of school-based health interventions [29, 30].

Following capacity building in our RCP program, more than 500 AYLHIV had their HIV status disclosed at schools in two counties in Kenya with high HIV prevalence. Similarly, a school-based teacher led initiative in Uganda contributed to a more open, welcoming, stigma-free environment that promoted stronger HIV openness between teachers and pupils; and in combination with multi-faceted activities such as edutainment and dance/dramas at school assemblies, enabled AYLHIV to disclose their status at school [31]. A qualitative study from Kisumu, Kenya reported that fear of stigma prevented the disclosure of HIV status to school staff among AYLHIV and led to higher rates of disengagement from care [16]. Our school-based package, which provided capacity building to school personnel and SHC, contributed to creating an enabling environment for AYLHIV. Importantly, the rates of viral suppression among AYLHIV within the area of RCP implementation, were higher (87% for younger adolescents (10–14 years), 90% for older adolescents (15–19 years) compared to the overall rates of viral suppression among ALHIV (10–19 years) at 61% in Kenya nationally during the same time period [32].

In addition to supporting AYLHIV, the RCP school-based package also served as a platform to reaching wider school-based communities with HIV, stigma, and SRH education. Despite some controversy surrounding the significance of school-based support in preventing HIV and other STIs among adolescents [33, 34], school-based programs have been shown to successfully implement HIV education, prevention and testing activities in schools [35, 36]. Limited studies from sub-Saharan Africa and Asia highlight primarily qualitative insights into the anticipated positive role schools can play in contributing to the overall health, wellbeing and academic success of AYLHIV students through the provision of care and treatment support [37, 38]. Significant gaps remain in evidence regarding feasibility of effective school-based interventions to improve health outcomes among AYLHIV in resource-limited settings affected by the HIV epidemic [16, 23, 39]. Because of the dearth in literature on school-based support for AYLHIV, the potential and value of treatment and care support in school settings remain largely unrecognized.

The RCP model provides evidence to establishing and maintaining comprehensive support for care and treatment of AYLHIV in schools and forming bi-directional linkages with HCFs while enabling broader HIV supportive school environments. The high level of collaboration between partners and stakeholders was essential to the success of RCP at schools. MOH, MOE and TSC in Homa Bay and Turkana Counties were consistently engaged, along with other stakeholders including AYLHIV, and led the planning and implementation of the school-based HIV interventions. Their involvement and continued coordination and support assured the success of the program. Establishing and sustaining meaningful relationships between health and education sectors at local and regional levels is crucial to enabling the implementation of school-based activities to support AYLHIV and enhancing HIV and SRH education among staff and students.

Our study has several limitations. We evaluated programmatic implementation, did not conduct qualitative baseline assessments and did not have a control group for the prospective evaluation of the school interventions. Not all schools were able to implement all elements of the RCP school package (though the majority implemented ≥80% of the required activities), which makes it challenging to draw conclusions about the efficacy of the separate elements of our school-based package of interventions. We also did not collect individual patient level data to assess a change in the self-management skills among AYLHV; however, we did conduct focus groups discussions with AYLHIV as part of the RCP project with a separate qualitative analysis on self-reported barriers to care, which remains in progress. Our findings are specific to Kenya and might not be generalizable to other country contexts. Despite these limitations, our study provides important contributions to the gap in the literature reporting feasibility of implementing school-based HIV care and treatment support in schools located in high HIV epidemic burden resource-limited settings.

## Conclusions

We report the feasibility of implementing a package of school-based interventions to support AYLHIV within boarding schools in two counties with high HIV prevalence in Kenya. We achieved strong collaboration between regional healthcare and education authorities and stakeholders including AYLHIV and their caregivers. Implementation of the RCP school-based package of interventions resulted in improved capacity to support AYLHIV and increased the number of AYLHIV with a disclosed HIV status, with access to ART and who were supported through coordination of care within boarding schools. This study addresses a gap in the literature around the feasibility, sustainability and outcomes of school-based HIV care and treatment support within boarding schools in resource-limited settings heavily affected by the HIV epidemic. Further research of effective evidence-based interventions is needed to strengthen approaches that optimize HIV care and treatment and supportive environments in school settings for millions of AYLHIV globally.

## Supporting information

**S1 Appendix. Components of RCP activities within the SHC.** This file outlines components of RCP activities coordinated through the SHC concerning identification of LLHIV, linkages to, and retention in care within schools. The HIV-responsive school checklist outlines criteria consistent with characteristics of an HIV-responsive school.
(DOCX)

## Acknowledgments

We would like to thank all the adolescents and youth supporting and participating in the RCP. Specifically, we would like to thank the members of the AYPAG who dedicated their time and efforts to the design of the program and continue to inspire and support it. We extend our special thanks to the regional healthcare and education leaders, school and healthcare facilities staff of Homa Bay and Turkana Counties in Kenya, who have supported the design and implementation of the RCP.

## Author Contributions

**Conceptualization:** Judith Kose, Cosima Lenz, Fred Kiiru, Justine Jelagat Odionyi, Rose Otieno-Masaba, Solomon Leselewa, Natella Rakhmanina.

**Data curation:** Judith Kose, Cosima Lenz, Job Akuno, Rose Otieno-Masaba, Natella Rakhmanina.

**Formal analysis:** Job Akuno, Rose Otieno-Masaba, Godfrey Woelk, Natella Rakhmanina.

**Investigation:** Elizabeth A. Okoth.

**Methodology:** Pieter Leendert Fraaij.

**Project administration:** Fred Kiiru, Justine Jelagat Odionyi, Elizabeth A. Okoth, Solomon Leselewa, Natella Rakhmanina.

**Supervision:** Pieter Leendert Fraaij.

**Validation:** Pieter Leendert Fraaij.

**Writing – original draft:** Judith Kose, Cosima Lenz, Job Akuno, Natella Rakhmanina.

**Writing – review & editing:** Judith Kose, Cosima Lenz, Fred Kiiru, Justine Jelagat Odionyi, Rose Otieno-Masaba, Elizabeth A. Okoth, Godfrey Woelk, Solomon Leselewa, Natella Rakhmanina.

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
