## [Decision Letter · Decision Letter 0]

25 Jun 2021

PONE-D-21-12670

Engaging boarding schools to support adolescents living with HIV in Kenya

PLOS ONE

Dear Judith Kose,

Thank you for submitting your manuscript to PLOS ONE. After careful consideration, we feel that it has merit but does not fully meet PLOS ONE’s publication criteria as it currently stands. Therefore, we invite you to submit a revised version of the manuscript that addresses the points raised during the review process.

ACADEMIC EDITOR: Dear Judith Kose,

Your manuscript covers an important topic and therefore requires to be read wide and large. However, the current version requires some attention and they are well highlighted by reviewers.

1. Ensure your study title, goal, and specific objectives of this submission align across the manuscript.

2. Provide more information on the consent, disclosure of HIV, and potential for stigma among the participants through getting involved in this project

3. Revisit the methodology section and add missing details on assessments done and analyses. Furthermore your reporting style should ensure flow and easy communication of findings to the reader

4. Your discussion and conclusion must only address that which is provided in the results

5. Ensure you bring out the special gap in knowledge that your manuscript addresses - apart from telling us about this great project, why should anyone care? Who should care? What's the special significance?

We look forward to receiving your revised manuscript.

Kind regards,

Violet Naanyu, PhD

Academic Editor

PLOS ONE

Journal Requirements:

'This work was supported with funding by a grant from the Positive Action for Adolescents Programme by ViiV Healthcare UK Ltd, London, UK under the Positive Action for Adolescents Program. EGPAF’s adolescent and youth-focused programs in Kenya are supported by funding from United States Government (USG), ELMA Philanthropies, and ViiV Healthcare UK Ltd, London, UK. The funder did not participate in the implementation of the program or interpretation of the results as presented in this manuscript. The results do not necessarily represent the views of the funder.'

We note that you received funding from a commercial source: ViiV Healthcare UK Ltd

Additional Editor Comments (if provided):

Dear Judith Kose,

Your manuscript covers an important topic and therefore requires to be read wide and large. However, the current version requires some attention and they are well highlighted by reviewers.

1. Ensure your study title, goal and specific objectives of this submission align across the manuscript.

2. Provide more information the consent, disclosure and potential for stigma among the participants through getting involved in this project

3. Revisit the methodology section and add missing details on assessments done and analyses. Furthermore your reporting style should ensure flow and easy communication of findings to the reader

4. Your discussion and conclusion must only address that which is provided in the results

5. Ensure you bring out the special gap in knowledge that your manuscript addresses - apart from telling us about this great project, why should anyone care? Who should care? What's the special significance?

Reviewers' comments:

Reviewer's Responses to Questions

**Comments to the Author**

1. Is the manuscript technically sound, and do the data support the conclusions?

Reviewer #1: Partly

Reviewer #2: Yes

2. Has the statistical analysis been performed appropriately and rigorously? 

Reviewer #1: No

Reviewer #2: Yes

3. Have the authors made all data underlying the findings in their manuscript fully available?

Reviewer #1: Yes

Reviewer #2: Yes

4. Is the manuscript presented in an intelligible fashion and written in standard English?

Reviewer #1: Yes

Reviewer #2: Yes

5. Review Comments to the Author

Reviewer #1: KEY MESSAGE:

- The study is an important one and will contribute to the knowledge gap regarding adolescents living with HIV in the boarding school environment.

- The initial impression is that the authors wished to evaluate the impact of the RCP intervention on retention in care and treatment. However, on further reading, the real question being asked is whether one can implement an intervention to improve student-school-HCF interactions in boarding secondary schools.

- The initial aim is therefore not addressed by the study findings

- I highlight what needs to be done in the various sections of the manuscript to improve its message and contribution to the scientific community.

A. Introduction:

- The question the authors sought to answer has changed from the introduction through the results and conclusions.

- On page 3 of the manuscript:

“The overarching goal of RCP is to increase linkages to and retention in care and treatment among AYLHIV aged 10-24 years. Implemented in coordination with the Kenya Ministry of Education (MOE) and Ministry of Health (MOH), the RCP school interventions included enhancing partnerships between boarding schools and RCP healthcare facilities (HCFs) while building capacity of schools to support AYLHIV.”

This does not seem to be captured in the results and conclusions of the manuscript. It however appears the main question was whether the intervention can be implemented.

Suggestion for improvement: Focus your introduction and aim both in abstract and main text on the intervention in the two counties and that is an offshoot of the main RCP school interventions.

- The context of the study in the introduction is clear.

B. Methods/Study design/Statistics:

- Well done.

- Few concerns that if addressed will make it more readable.

- Waiver for consent was requested and granted because of the minimal risk intervention. Did you consider the psychological consequences of inadvertent disclosure for those in school without complete disclosure about their HIV status?

- If so, it is not clear from the description how that was addressed during the study implementation.

- What of the stigma related to classmates realizing their HIV status?

- How was this mitigated against because it could have far reaching effects on the adolescent? If this was addressed, please outline how?

- Also, it would help to know if all schools were selected or if not, the method used in identifying the intervention schools.

- You mention as a limitation that you did not have non-intervention schools to compare with. Any reason why you did not consider intervention and non-intervention schools.

- It is not clear how improved linkage between Schools and HCT was assessed. In other words, if a child has a primary HCF from outside the catchment of the study, were they transferred to the county HCF?

- DOT: How was the improvement in this assessed? And how was it practically done?

- Counselors: How was their improved skill/improved psychosocial support assessed?

- Training curriculum for teachers: You used the HCP APOC curriculum. This is not been purposed for secondary school teachers. How did you modify it to enable the education given to the schools to be dependable?

Suggested improvements: Explain each item above, if considered how it was addressed?

- Outcomes: Your main outcomes seem to increased numbers of Secondary school-HCF units, increased AYLWH who were paired to HCF, number of teachers and counsellors trained on HIV care.

- I would have expected additional outcomes which were readily available to you to be presented in methods and results. These include adherence to medications, viral suppression (which comes in the discussion although missing in results), and psychological well-being using standard tools for assessment.

Suggested improvement: If you have contact of the individual AYLWH you can improve the manuscript by looking at baseline data on adherence to medications, viral suppression, and psychological well-being on these participants and including it.

- Outcome changes: The increased numbers from school-HCF units, teachers, counselors etc were never subjected to inferential statistics.

Suggested improvement: If you have pre and post-intervention numbers, you some inferential statistics can be done. However, the best approach would have been using a non-intervention group.

C. Results, Tables and Figures:

- -Overall impression: This is fairly well done. However some data is mentioned as percentages in the text but no raw number accompany that data.

- Examples:

“Overall, throughout RCP implementation, the number of schools with competent guidance and counselling teachers increased by 19.4%, the number of schools with teachers trained on caring and supporting AYLHIV increased by 99%, and the number of schools linked to a HCFs increased by 62%, compared to baseline”.

Suggestion for improvement: The bolded numbers need to have raw number –for example from xx to xx (19.4%). Having some of the data in the tables and referring to them will improve readability.

D. Conclusions:

- Indicates there is potential to improve retention in care and reduction of stigma. However, the results do not have any outcomes for any of the potential heath indicators.

Suggested change: Indicate the specific parameters that improved although the improvements were not tested statistically to confirm significance.

E. References and Appendices:

- Fairly well done

Thank you.

Reviewer #2: The paper is well written and has reviewed the latest literature in the area area related to the study. The methodology for implementing the intervention, as well as the methods of data analysis have been presented in a logical manner. However, while the paper has demonstrated the relevance of its findings in interventions that address HIV/AIDS and related challenges especially among the school-going youths, there has been little effort to demonstrate the scholarly contributions of the study. How does this study contribute to the field of sociology or related broad subject areas? What scholarly (academic) gaps was it designed to address and what new knowledge has the study contributed?

6. PLOS authors have the option to publish the peer review history of their article (what does this mean?). If published, this will include your full peer review and any attached files.

Reviewer #1: **Yes: **Winstone Nyandiko

Reviewer #2: **Yes: **Abraham Kiprop Mulwo

---

## [Author Response · Author response to Decision Letter 0]

10 Aug 2021

Reviewers’ Comments

Your manuscript covers an important topic and therefore requires to be read wide and large. However, the current version requires some attention, and they are well highlighted by reviewers.

1. Ensure your study title, goal, and specific objectives of this submission align across the manuscript.

Thank you for this comment. We have aligned study title, goal, and specific objectives across the manuscript. 

2. Provide more information on the consent, disclosure of HIV, and potential for stigma among the participants through getting involved in this project

We have added details on the consent, disclosure of HIV status and on the stigma considerations and measures to address it among the participants to the manuscript under Methods [lines 170-185; 166-190] and Discussion [lines 271-280].

3. Revisit the methodology section and add missing details on assessments done and analyses. Furthermore, your reporting style should ensure flow and easy communication of findings to the reader

As suggested, we revised methodology section and added more details on the assessments and data analysis [lines 218-228]. We have also adjusted our reporting style to ensure better flow and easy communication of the findings to the reader. As part of this revision, we have removed Figure 3 to be able to focus on the results of the whole project during the implementation period. 

4. Your discussion and conclusion must only address that which is provided in the results

As suggested, we revised the manuscript to make sure that the Discussion and Conclusion sessions only address our results. 

5. Ensure you bring out the special gap in knowledge that your manuscript addresses - apart from telling us about this great project, why should anyone care? Who should care? What's the special significance?

As suggested, we provided the evidence and highlighted the gaps and significance of our findings for the readers [lines 258-270; 289-300]. 

Reviewer #1

KEY MESSAGE: 

- The study is an important one and will contribute to the knowledge gap regarding adolescents living with HIV in the boarding school environment. 

- The initial impression is that the authors wished to evaluate the impact of the RCP intervention on retention in care and treatment. However, on further reading, the real question being asked is whether one can implement an intervention to improve student-school-HCF interactions in boarding secondary schools.

- The initial aim is therefore not addressed by the study findings

- I highlight what needs to be done in the various sections of the manuscript to improve its message and contribution to the scientific community.

We thank Reviewer 1 for these comments. Indeed, we describe the program with multiple components, some of them have been published already and are cited. We did focus this analysis on the school interventions only and have clarified that the initial aim of this analysis was for the analysis of the feasibility of school interventions [lines 76-87]; the healthcare facility interventions have been described in a separate published manuscript. We made sure to reflect this focus in various sections of the manuscript as suggested by the reviewer. 

A. Introduction: 

- The question the authors sought to answer has changed from the introduction through the results and conclusions. 

- On page 3 of the manuscript:

“The overarching goal of RCP is to increase linkages to and retention in care and treatment among AYLHIV aged 10-24 years. Implemented in coordination with the Kenya Ministry of Education (MOE) and Ministry of Health (MOH), the RCP school interventions included enhancing partnerships between boarding schools and RCP healthcare facilities (HCFs) while building capacity of schools to support AYLHIV.”

This does not seem to be captured in the results and conclusions of the manuscript. It however appears the main question was whether the intervention can be implemented. 

Suggestion for improvement: Focus your introduction and aim both in abstract and main text on the intervention in the two counties and that is an offshoot of the main RCP school interventions. 

- The context of the study in the introduction is clear.

As pointed out by the reviewer, we describe the program with multiple components, some of them have been published already and we focus this analysis on the school interventions only. We have clarified that initial aim of this analysis was to focus on school interventions [lines 85-87; 91-93]. We made sure to reflect this focus in the Introduction and clarified that the separate analysis was conducted for all components of the RCP package. 

B. Methods/Study design/Statistics:

- Well done.

- Few concerns that if addressed will make it more readable.

- Waiver for consent was requested and granted because of the minimal risk intervention. Did you consider the psychological consequences of inadvertent disclosure for those in school without complete disclosure about their HIV status? 

- If so, it is not clear from the description how that was addressed during the study implementation. 

- What of the stigma related to classmates realizing their HIV status? 

- How was this mitigated against because it could have far reaching effects on the adolescent? If this was addressed, please outline how? 

We thank the reviewer for pointing out this important area for clarification. We have clarified further our disclosure interventions. The disclosure of HIV status was offered to AYLHIV and/or their disclosed caregivers. AHA, SHC and selected school staff were trained on supporting two types of disclosure of HIV status: a) by AYLHIV to the school staff, caregivers and peers where applicable; and b) by caregivers to the school staff and AHA and undisclosed AYLHIV where applicable. AHA, SHC and school staff made the wider school community (including PA and learners) aware that disclosure support was available and emphasized the confidentiality and discretion of the process. RCP school staff, SCH and AHA regularly reached out to the school community reaffirming the available support for disclosure of HIV status. Every effort was made to provide disclosure support in a way that would minimize accidental disclosure and prevent internal and external stigma. Accidental or inadvertent disclosure was not reported in our study. Within the wider school community messaging about stigma, its harms and messaging about living positive with HIV were disseminated as part of RCP implementation. We clarified all the disclosure process, measures, and stigma mitigation in the manuscript [lines 171-190]. 

- Also, it would help to know if all schools were selected or if not, the method used in identifying the intervention schools.

 To address this point, we have clarified the process of school selection in Methods [lines 101-113].

- You mention as a limitation that you did not have non-intervention schools to compare with. Any reason why you did not consider intervention and non-intervention schools. 

The reviewer is correct, we did not have control group of non-intervention schools and did describe this as a limitation of our study. RCP was a programmatic intervention and not a prospective trial of new interventions, therefore including control group of non-intervention schools while being desirable, was not feasible. 

- It is not clear how improved linkage between Schools and HCT was assessed. In other words, if a child has a primary HCF from outside the catchment of the study, were they transferred to the county HCF? 

Thank you for this comment. We had only few AYLHIV (10) who had their HCF outside of the catchment areas. Those were not transferred to RCP HCF, but they received school-based care and treatment support and PSS support. The linkage between the schools and HCF was assessed through documenting the mutual referral/communication processes, point of contact staff assigned at healthcare facilities and schools, documentation of the HCF within the school health records for the disclosed AYLHIV and documentation of school-based support (access to ART, DOT and PSS) within HCF records. Programmatic monitoring and evaluation visits were used to verify the bi-directional connections between HCF and schools with random client reviews. We included all these clarifications in the manuscript. [lines 200-201; 218-223; 248-249]

- DOT: How was the improvement in this assessed? And how was it practically done?

We have clarified our DOT process in detail including practical aspects under the Methods [lines 190-193]. 

- Counselors: How was their improved skill/improved psychosocial support assessed? 

We did not conduct a systematic assessment of the skills among AYLHV. We report on the strengthening of the PSS service and clarified this in the manuscript [lines 160-163; 205-211; 245-247]. 

- Training curriculum for teachers: You used the HCP APOC curriculum. This is not been purposed for secondary school teachers. How did you modify it to enable the education given to the schools to be dependable? 

Suggested improvements: Explain each item above, if considered how it was addressed? 

We have clarified in the Methods that the training for teachers and school support personnel was conducted using an EGPAF developed curriculum adapted using language from the national Adolescent Package of Care (APOC) training. Our training curricula focused on enhancing school personnel understanding of the current management of HIV care, antiretroviral treatment (ART) and support needs for AYLHIV with a focus on disclosure of HIV, stigma, treatment adherence support, ART storage, psychosocial support and coordination of care [lines 145-150; 160-163]. 

- Outcomes: Your main outcomes seem to increased numbers of Secondary school-HCF units, increased AYLWH who were paired to HCF, number of teachers and counsellors trained on HIV care. 

- I would have expected additional outcomes which were readily available to you to be presented in methods and results. These include adherence to medications, viral suppression (which comes in the discussion although missing in results), and psychological well-being using standard tools for assessment. 

Suggested improvement: If you have contact of the individual AYLWH you can improve the manuscript by looking at baseline data on adherence to medications, viral suppression, and psychological well-being on these participants and including it.

Thank you for these comments. Collecting data on adherence and psychological well-being among AYLHIV requires additional resources and was not included in the scope of our evaluation. For viral suppression, we have reported rates of viral suppression from our programmatic data under the Discussion [lines 280-284] and now extended it under Results [lines 250-253]. 

- Outcome changes: The increased numbers from school-HCF units, teachers, counselors etc were never subjected to inferential statistics. 

Suggested improvement: If you have pre- and post-intervention numbers, you some inferential statistics can be done. However, the best approach would have been using a non-intervention group. 

As noted by the reviewer, we report post-intervention numbers. For the baseline assessment, we conducted a mapping analysis prior to the start of the project but did not include the qualitative evaluation. As clarified above, we did not have a control non-intervention group for this program implementation project. 

C. Results, Tables and Figures: 

- -Overall impression: This is fairly well done. However, some data is mentioned as percentages in the text, but no raw number accompany that data. 

- Examples: 

“Overall, throughout RCP implementation, the number of schools with competent guidance and counselling teachers increased by 19.4%, the number of schools with teachers trained on caring and supporting AYLHIV increased by 99%, and the number of schools linked to a HCFs increased by 62%, compared to baseline”. 

Suggestion for improvement: The bolded numbers need to have raw number –for example from xx to xx (19.4%). Having some of the data in the tables and referring to them will improve readability. 

Thank you for this suggestion. The referenced text was related to Figure 3 which reported data from project year two only. To avoid confusion and to better focus on the results of the project overall, we have removed Figure 3 and relevant text from the manuscript. 

D. Conclusions:

- Indicates there is potential to improve retention in care and reduction of stigma. However, the results do not have any outcomes for any of the potential heath indicators. 

Suggested change: Indicate the specific parameters that improved although the improvements were not tested statistically to confirm significance.

We have realigned our Conclusions [lines 323-329] to reflect the descriptive statistical findings of our evaluation study and also clarified this within the Discussion. 

E. References and Appendices:

- Fairly well done

Thank you, based on our revisions we have slightly expanded and also updated the references [lines 431-433; 587-590; 591-594]. 

Reviewer #2 

The paper is well written and has reviewed the latest literature in the area related to the study. The methodology for implementing the intervention, as well as the methods of data analysis have been presented in a logical manner. However, while the paper has demonstrated the relevance of its findings in interventions that address HIV/AIDS and related challenges especially among the school-going youths, there has been little effort to demonstrate the scholarly contributions of the study. How does this study contribute to the field of sociology or related broad subject areas? What scholarly (academic) gaps was it designed to address and what new knowledge has the study contributed?

We have clarified the major gap in the subject area, as very limited data on the effective school interventions to support AYLHIV. Specifically, we added the citation from WHO: “The WHO acknowledges the critical opportunity in linking schools and community services to support an integrated approach to health along with the need for capacitation and active engagement of teachers and school staff in school-based health programs.[26]” [lines 258-2610

We clarified further that our project was designed to improve disclosure of HIV status at schools to enable care and treatment support. Considering the dearth in literature exploring evidence-based school-based support for AYLHIV, the potential and value of care and treatment support in school environments needs to be recognized. [lines 271-280]

We also emphasized that our study contributes to the gap in the literature addressing the feasibility of implementing school-based HIV care and treatment support in high HIV epidemic burden, resource-limited settings. In our study, building capacity of school staff, parents, adolescents, and youth improved availability and provision of coordinated care within schools including adherence support, counselling, access, and storage of medication for AYLHIV, while expanding the HIV prevention and SRH education platform for the broader school community. [lines 286-300]

We have also added one more co-author who participated in the data analysis and contributed to the critical review of manuscript with reviewers’ feedback. Please, let us know if any additional information or clarifications are required. 

Thank you for your consideration of our revised manuscript.

---

## [Decision Letter · Decision Letter 1]

8 Nov 2021

Supporting adolescents living with HIV within boarding schools in Kenya

PONE-D-21-12670R1

Dear Dr. Judith Kose,

We’re pleased to inform you that your manuscript has been judged scientifically suitable for publication and will be formally accepted for publication once it meets all outstanding technical requirements.

Kind regards,

Violet Naanyu, PhD

Academic Editor

PLOS ONE

Reviewer's Responses to Questions

**Comments to the Author**

1. If the authors have adequately addressed your comments raised in a previous round of review and you feel that this manuscript is now acceptable for publication, you may indicate that here to bypass the “Comments to the Author” section, enter your conflict of interest statement in the “Confidential to Editor” section, and submit your "Accept" recommendation.

Reviewer #1: All comments have been addressed

Reviewer #2: All comments have been addressed

2. Is the manuscript technically sound, and do the data support the conclusions?

Reviewer #1: Yes

Reviewer #2: Yes

3. Has the statistical analysis been performed appropriately and rigorously? 

Reviewer #1: Yes

Reviewer #2: Yes

4. Have the authors made all data underlying the findings in their manuscript fully available?

Reviewer #1: Yes

Reviewer #2: Yes

5. Is the manuscript presented in an intelligible fashion and written in standard English?

Reviewer #1: Yes

Reviewer #2: Yes

6. Review Comments to the Author

Reviewer #1: My concerns and suggested edits have been done comprehensively. The authors have satisfactory addressed my comments, questions and suggestions.

Reviewer #2: Concerns that I had raised have been sufficiently addressed. the article may now be processed for publication

7. PLOS authors have the option to publish the peer review history of their article (what does this mean?). If published, this will include your full peer review and any attached files.

Reviewer #1: **Yes: **Winstone Nyandiko, Professor of Pediatrics, Moi University

Reviewer #2: **Yes: **DR. Abraham Kiprop Mulwo

---

## [Editor Report · Acceptance letter]

22 Nov 2021

PONE-D-21-12670R1 

Supporting adolescents living with HIV within boarding schools in Kenya 

Dear Dr. Kose:

I'm pleased to inform you that your manuscript has been deemed suitable for publication in PLOS ONE. Congratulations! Your manuscript is now with our production department. 

Kind regards, 

on behalf of

Prof. Violet Naanyu 

Academic Editor

PLOS ONE